# Ultrafast phonon-mediated dephasing of color centers in hexagonal boron nitride probed by electron beams

M. Taleb[1,2], P. H. Bittorf[1], M. Black [1], M. Hentschel [3], W. Sigle[4], B. Haas [5], C. Koch [5], P. A. van Aken [4], H. Giessen [3] & N. Talebi [1,2] ✉

Defect centers in hexagonal boron nitride (hBN) have been extensively studied as room-temperature single-photon sources. The electronic structure of these defects exhibits strong coupling to phonons, as evidenced by the observation of phonon sidebands in both photoluminescence and cathodoluminescence spectra. However, the dynamics of the electron-phonon coupling as well as phonon-mediated dephasing of the color centers in hBN remain unexplored. Here, we apply a novel time-resolved CL spectroscopy technique to explore the population decay to phonon states and the dephasing time $T_2$ with sub-femtosecond time resolution. We demonstrate an ultrafast dephasing time of only 200 fs and a radiative decay of about 585 fs at room temperature, in contrast with all-optical time-resolved photoluminescence techniques that report a decay of a few nanoseconds. This behavior is attributed to efficient electron-beam excitation of coherent phonon-polaritons in hBN, resulting in faster dephasing of electronic transitions. Our results demonstrate the capability of our sequential cathodoluminescence spectroscopy technique to probe the ultrafast dephasing time of single emitters in quantum materials with 1.5 fs time resolution, heralding access to quantum-path interferences in single emitters coupled to their complex environment.

Van der Waals materials have been extensively studied due to their fascinating multi-physics functionalities. They provide a platform for strongly correlated materials[1,2] and a landscape for polariton physics[3]. In particular, hexagonal boron nitride (hBN) appears to be a strong candidate for the various forms of light-matter interactions. In the infrared, hBN hosts phonon polaritons[4–6], while in the visible and ultraviolet, defect centers in hBN appear as room-temperature single-photon emitters[7–9]. Several forms of defect centers, emitting number-state photons with their wavelengths spanning the entire visible to ultraviolet range have been reported and studied using photoluminescence (PL)[10,11] and cathodoluminescence (CL)[9,12] spectroscopy.

The emitters in hBN also exhibit exceptional spin[13,14] and electro-optical[15,16] properties.

Room-temperature photon emission from defect centers in other materials, such as diamond[17,18] and GaN[19], has been extensively researched in parallel and has established itself as a paradigm for quantum-optics-based technologies[20]. However, equivalent emitters in a thin van der Waals material with a refractive index lower than that of diamond allow for an efficient implementation of the emitters in solid-state quantum networks and devices, enabling a broad range of applications for future quantum technologies[21–23]. Defect centers in hBN, therefore, manifest themselves as such a candidate.

[1]Institute of Experimental and Applied Physics, Kiel University, Kiel, Germany. [2]Kiel Nano, Surface and Interface Science KiNSIS, Kiel University, Kiel, Germany. [3]4th Physics Institute and Research Center SCoPE, University of Stuttgart, Stuttgart, Germany. [4]Stuttgart Center for Electron Microscopy, Max Planck Institute for Solid State Research, Stuttgart, Germany. [5]Department of Physics & Center for the Science of Materials Berlin (CSMB), Humboldt-Universität zu Berlin, Berlin, Germany. ✉e-mail: talebi@physik.uni-kiel.de

The photophysics of the defect centers in hBN is characterized by strong coupling of the emitters to phonons[24–26]. The excitation of phonons reduces the dephasing time of electronic transitions and represents a limit for the realization of Fourier-transform-limited emitters. Therefore, the decoupling of quantum emitters trapped between hBN layers from in-plane phonon excitations has been discussed as a mechanism to achieve Fourier-transform-limited transitions[27,28]. Despite all these efforts, a direct probing of the phonon-mediated decoherence mechanisms and dephasing of single hBN emitters has remained unexplored, partly due to the limitations of all-optical techniques to probe femtosecond dynamics at deep subwavelength spatial dimensions.

In contrast to light, electron beams can be focused to sub-nanometer spot sizes and excite single defect centers in solid-state materials, such as hBN[29] and diamond[30]. In particular, several forms of defect centers have been studied using CL spectroscopy[12,31–33] that emit in the entire visible range. However, the photophysics dynamics of the emitters, such as the phonon-mediated dephasing of single emitters, is still unexplored, even with electron beams.

Here, we use a recently developed phase-locked photon–electron spectroscopy technique based on sequential CL spectroscopy to unravel the phonon-mediated dephasing time of quantum emitters[34,35]. Using a broadband metamaterial-based electron-driven photon source[36,37] (EDPHS), which emits sub-cycle photons with a collimated spatial profile and temporal distribution of 1.4 fs, we generate a coherent superposition of phonon states. Therefore, the CL emission from quantum emitters after the interaction with the EDPHS radiation exhibits coherent and incoherent contributions. Moreover, distinguishing the dynamics of coherent and incoherent CL emission from the delay between the electron and EDPHS photons exciting the sample leads to the determination of both the population relaxation ($T_1$) and dephasing ($T_2$) time scales of the emitters interacting with electron beams, which are of the order of $T_1 = 200$ fs and $T_2 = 580$ fs, which is significantly smaller than the reports based on all-optical characterization techniques. We provide a theoretical model based on

a master equation for single emitters coupled to both EDPHS light and electron beams, which agrees well with our experimental results, and demonstrates the important aspect of incoherent electron excitation in understanding the physics of the interactions and the increase in the decay time.

Our work not only provides valuable insights into the photophysics of hBN emitters coupled to phonons, but also enables a deep understanding of the mechanisms of the radiation from emitters excited with electron beams. Moreover, it demonstrates the unique ability of our CL technique to couple to deep subwavelength emitters, paving the way for future applications in probing the dynamics of quantum emitters implemented in integrated photonic networks and solid-state quantum devices.

## Results
### Time-resolved cathodoluminescence spectroscopy of hBN defect centers

The hBN defects employed in our experiments are formed by liquid exfoliation of pristine hBN flakes from high-quality crystals onto holey carbon transmission electron microscopy grids. We have analyzed various liquids to identify quantum emitters that remain stable under intense electron beam illumination, as detailed in the Methods section. In particular, we found that the exploitation of isopropanol leads to scarcely-positioned stable emitters in thin hBN flakes. The emission wavelength of most stable emitters is centered at $\lambda_1 = 880$ nm, that is followed by two phonon sidebands at the wavelengths of $\lambda_2 = 797$ nm and $\lambda_3 = 670$ nm (Fig. 1a, b). The electronic transitions in hBN defect centers are strongly coupled to phonon excitations, which form a quantum ladder in the potential landscape of the atomic defect[26] and lead to sequential relaxation of the population generated by electron beams, and subsequent emission of photons in an incoherent manner (Fig. 1a). This fact is confirmed by the CL spectra obtained from the defect centers at room temperature (Fig. 1b).

A moving electron exciting the defect statistically populates the quantum system into higher order states, mainly enabled by bulk

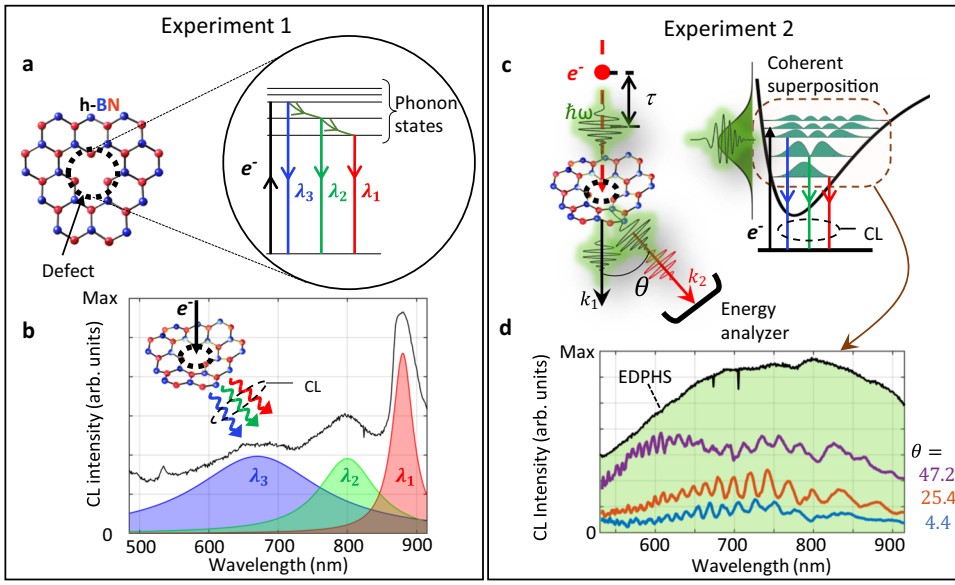

**Fig. 1 | Phonon-mediated electronic transitions and coherence.** Experiment 1: **a** (Left) Schematic of an atomic defect in hBN and its energy diagram (right), representing a two-level electronic system coupled to higher energy phonon states. A swift electron interacting with the defect incoherently populates the defect to its higher energy states. Sequential transitions to lower energy phonon states, followed by an electronic transition to the ground state, emit photons of different wavelengths, resulting in the observed distinct peaks shown in the CL spectrum (**b**). Experiment 2: **c** An ultrabroadband and coherent chirped optical pulse, emitted

from the EDPHS, generates a coherent superposition of the phonon states. After the electron interaction, the CL emission from the initially superpositioned system has two contributions: a coherent and incoherent CL radiation. The coherent CL emission is prominently recovered by acquiring the momentum-resolved CL radiation along specific angular ranges (angle-resolved CL mapping). **d** Spectral interferences in the acquired CL intensity for a defect center already prepared in a superposition, at the depicted polar angles $\theta$. The EDPHS spectrum is indicated by the green-shaded area. Source data are provided as a Source Data file.

plasmons[38], phonons[39,40], or cascaded interaction of secondary electrons with defects[41], without generating a coherent superposition of the states. Incoherent light generation from single defects is associated with the generation of light in number states, where the expectation value of the field operator ideally vanishes, similar to PL experiments with defect centers[42–44]. This is fundamentally different from the interaction of quantum systems with coherent light pulses, which generate a coherent superposition of the states depending on the frequency and duration of the pulses[45]. The timescale within which the generated coherence remains in the system, i.e., the dephasing time, is crucial for the realization of Fourier-transform-limited single-photon sources, and interferometry techniques based on single-photon emitters[46].

In order to better comprehend the dynamics of the quantum system interacting with the electron beams, we use here a quantum master equation that is particularly suitable for modeling the incoherent generation of photons, recast as[42]

$$\frac{d\hat{\rho}}{dt} = -\frac{i}{\hbar}[\hat{H}, \hat{\rho}] + \hat{D}_{\text{rad}}\hat{\rho} + \hat{D}_{\text{ex}}\hat{\rho} \qquad (1)$$

where $\hat{\rho}$ is the density matrix and $\hat{H}$ is the system Hamiltonian. $\hat{D}_{\text{rad}}$ and $\hat{D}_{\text{ex}}$ are the Lindblad operators associated with the spontaneous emission and electron beam excitations, respectively (see Supplementary Note 1). First, without interaction with external light, within which the Hamiltonian remains as $\sum_{n=1}^{N}\hbar\omega_n|n\rangle\langle n|$, the dynamics of the diagonal terms of the density matrix are decoupled from the off-diagonal terms, representing the generation of an incoherent population due to the interaction with electron beams. Second, the generated CL emission from the system is modeled with the spectral density function, linked with the expectation value of the photon number operator ($\sigma^+\sigma^-$) in the frequency domain and is derived as

$$S(\omega) = \text{Re}\int_0^{+\infty} d\tau\, e^{-i\omega\tau}\int_{-\infty}^{+\infty} dt \sum_{m<n}\text{tr}\{\hat{\sigma}_{mn}^+(t)\hat{\sigma}_{mn}^-(\tau-t)\rho(t)\} \qquad (2)$$

where $\hat{\sigma}_{mn}^+$ and $\hat{\sigma}_{mn}^-$ are the creation and annihilation operators associated with the transitions from the $m$th to the $n$th state. Within the weak interaction regime, $S(\omega)$ is related to the Fourier transform of the diagonal terms of the density matrix ($\rho_{nn}$ with $n>1$), thus relating the incoherent CL emission to the population relaxation. Particularly, we notice, that in order to model the CL spectra obtained experimentally here, the consideration of eight quantum states is required (see Supplementary Note 1), mainly due to the anharmonic nature of the potential landscape, which leads to the free induction decay[47] and an additional broadening of the CL phonon peaks. The phonon quantum states initiate from a molecular-like system with a densely packed and unequally spaced quantum ladder for phonons with their transition wavelengths to the ground state positioned between 550 nm and 797 nm (Supplementary Figs. 1–3).

To probe the dephasing time of phonon states, we use an EDPHS as an internal radiation source inside a scanning electron microscope (SEM) to generate optical pulses that are phase-locked to the near-field of the moving electron[36]. Our EDPHS is fabricated using focused ion milling to create a pattern of distributed nanopinholes in a gold thin film, positioned on top of a $Si_3N_4$ membrane (see Supplementary Note 2 and Supplementary Figs. 4 and 5). The position of the nanopinholes is pre-designed to enable a collimated beam profile[34], for an electron interacting with the EDPHS in the central region of the structure. The emission from the EDPHS is ultrabroadband, covering the spectral range from 560 nm to 940 nm (Fig. 1d). This ultrabroadband and coherent emission, generates a coherent superposition of phonon states. The delay between the EDPHS radiation and the swift electron interacting with the sample is controlled via a piezo stage by

changing the distance between the sample and the EDPHS as $\tau = L(v_{\text{el}}^{-1} - c^{-1})$. Here, $L$ is the distance between the sample and the EDPHS, $v_{\text{el}}$ is the group velocity of the electron in the vacuum, and $c$ is the speed of light.

Therefore, the CL emission from the incident electron beam interacting with the coherently superposed quantum states now has two counterparts: a coherent part and an incoherent part. The coherent radiation from the emission centers is distinguished from the incoherent emission by performing interferometry with the generated CL light from the sample and the EDPHS (See Supplementary Fig. 6). The CL emission naturally interferes with the coherent EDPHS radiation, forming spectral interference fringes, that is revealed by decomposing the total emission into its different angular components (Fig. 1d). Moreover, as we will show below, the visibility of the interference fringes changes by changing the delay between the incoming electron and the EDPHS radiation, which allows us to retrieve the dephasing time of the generated phonon superposition. In this case, the Hamiltonian of the system interacting with the EDPHS radiation changes as $\sum_{n=1}^{N}\hbar\omega_n|n\rangle\langle n| - \hat{\mu}\cdot\vec{E}(t)$, where $\vec{E}(t)$ is the electric field associated with the EDPHS radiation and $\hat{\mu}$ is the dipole transfer matrix of the system. The coherent CL radiation arises from the induced coherent polarization in the system, modeled as $P(t,\tau) = \text{tr}\{\hat{\mu}\hat{\rho}(\mathbf{t},\tau)\}$, where $t$ is the elapsed time and $\tau$ is the delay between the EDPHS and electron excitations. The polarization, unlike the expectation value of the photon number operator (Eq. (2)), is related to the off-diagonal elements of the density matrix, which is now nonzero due to the interaction with the EDPHS light. Therefore, its dynamic is related to the dephasing time of the system, as will be shown below.

## Defect centers in hBN

The nature of the defect centers in hBN is widely debated. While almost all experiments clearly demonstrate the strong interaction between phonons and electrons, the nature of the atomic structure of the defect is still not fully understood. Here, we perform CL spectroscopy, high-resolution transmission electron microscopy (HRTEM), and low-energy electron energy-loss spectroscopy (EELS) to shed light on the nature of the defects and phonon excitations.

Our CL spectroscopy measurements clearly show the excitation of two types of defects, whose emission wavelengths are strongly thickness dependent. In the region of interest where we perform our time-resolved CL spectroscopy measurements, we denote the excitation of electronic transitions, with the emission wavelength at $\lambda_1 = 880$ nm, followed by two phonon sidebands (Fig. 2a). The defect distribution is resolved by performing spectral imaging, where we plot the CL intensity corresponding to the emissions at $\lambda_1 = 880$ nm, $\lambda_2 = 797$ nm, and $\lambda_3 = 670$ nm versus the scan position (Fig. 2b). The spectral image associated with the electronic transitions at $\lambda_1 = 880$ nm, shows a scattered distribution of the emitters. However, the first phonon peak is homogeneously distributed within the thicker region of the flake at some distance from the edge, and the second phonon transition is more localized at the edge.

In order to explore the phonon excitations in our hBN flakes in more detail, we perform low-energy EELS (LE-EELS) with a Nion electron microscope[48,49] (Fig. 2c, d). Our LE-EELS measurements show the excitation of three distinct and closely spaced phonon peaks, at the energies of $E_1 = 157$ meV, $E_2 = 169$ meV, and $E_3 = 186$ meV (Fig. 2c). The difference between the CL peaks at $\lambda_1$ and $\lambda_2$, when translated to the energy scale, agrees well with the phonon resonances revealed by the LE-EELS measurements. Moreover, the distribution of the phonon resonances at the resonant peak $E_2 = 169$ meV is more localized along the edges, while the other resonances are more homogeneously distributed inside the bulk, which agrees well with the distribution of the phonon sidebands revealed by CL spectral imaging. The highly non-

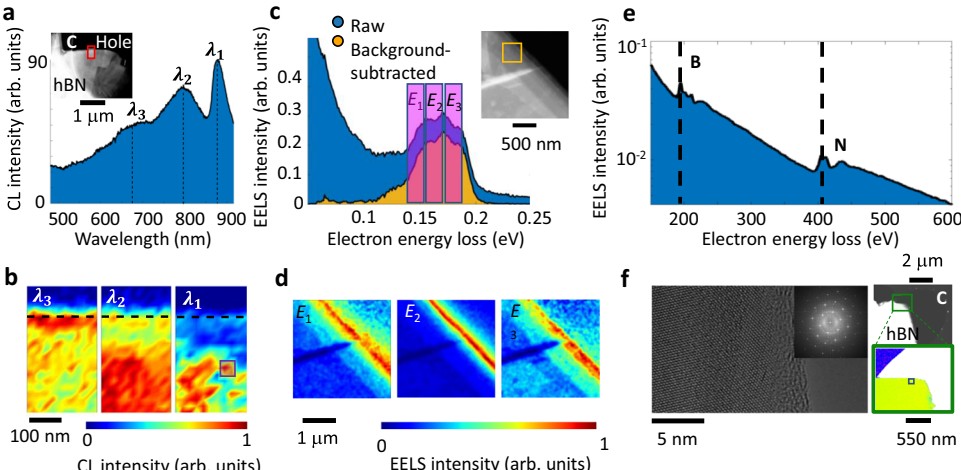

**Fig. 2 | Hyperspectral CL and EELS maps and structural analysis of hBN flakes. a** CL spectrum of the hBN flake integrated over the red box, featuring three different emission energies, marked by $\lambda_1$, $\lambda_2$, and $\lambda_3$. The inset depicts an SEM image of the measured hBN flake, which is placed on a holey carbon film by liquid exfoliation. The red area marks the measurement area. **b** Hyperspectral CL images of the marked area for the three spectral peaks indicated. The purple box shows the position at which the results in Figs. 3 and 4 are acquired. **c** Low-loss electron energy loss spectra of the flake showing three distinguished phonon peaks. The inset shows the transmission electron microscopy image of the flake. **d** Scanning EELS images of the inset figure in (**c**), integrated along the highlighted energy regions $E_1$, $E_2$, and $E_3$, showing the spatial distribution of the phonon resonances.

**e** High-energy EELS measurement of the hBN flake showing peaks at the energies associated with the boron and nitrogen K-edge transitions, but no peak for carbon. The results are obtained from the region shown by a black box in (**f**) right bottom panel. **f** (Left) HRTEM image of an hBN flake. The larger image shows the atomic structure of the flake, indicating the presence of some defects in the atomic lattice. The inset shows the Fourier transform of the image. (Right) The transmission electron microscopy image of the measured area and the resulting color-coded image representing the atomic composition of hBN flake on holey carbon. Here, boron and nitride are yellow and green, respectively, while carbon is blue and oxygen is red.

localized behavior of the phonon resonances as well as their strong thickness dependence suggest the excitation of phonon polaritons. hBN flakes are extensively studied within the Reststrahlen lower and upper bands, and electron beams couple particularly strongly to coherent hyperbolic phonon polaritons in hBN[50]. The energy range of the phonons in our flakes is within the upper Reststrahlen band of hBN (169–200 meV), which is sandwiched between the transverse-optical and longitudinal-optical phonon energies, and is expected to couple effectively to polaritons as well.

In addition, to better explore the nature of the electronic transitions, and in particular to understand whether the defects are due to external atomic impurities such as carbon[51,52], we performed analytical high-energy EELS (HE-EELS) (Fig. 2e). First, the HRTEM image of the flake shows the high-quality single-crystal nature of the flakes (Fig. 2f). The Fourier-transformed image as displayed in the inset, better represents the crystallinity of the flakes. Moreover, the high-energy electron energy-loss spectrum does not show any K-edge transition associated with the carbon ($E_C = 290$ eV) within the acquisition window considered here (Fig. 2e). Therefore, we rule out the excitation by an external carbon defect, especially since the density of the defects associated with the transitions is quite high in the flakes.

In addition to the defects studied above, some of our liquid-exfoliated flakes exhibit another class of defects, with the emission centered at the wavelength of $\lambda = 570$ nm (Supplementary Note 4 and Supplementary Figs. 7 and 8), indicating a double-peak nature, which is further revealed by the PL spectroscopy measurements (Supplementary Fig. 9). However, the peak centered at the energy of $\lambda = 880$ nm, is not visible in the PL spectra. By performing both CL and PL measurements on different flakes and at different positions, all of which show similar results, we conclude that the transition resonance at $\lambda = 880$ nm is dark in the PL measurements and carries a dipole oriented perpendicular to the plane of the flake, generated by boron vacancies[53], and thus perfectly couples to the electron-beam excitations. This claim is particularly supported by the better coupling of the radially polarized

light generated by the EDPHS to the phonon excitations (Fig. 1d and Fig. 3). The emission wavelength does not change when the kinetic energy of the electron beam and its current are varied (Supplementary Fig. 10), allowing us to rule out the generation of electron-beam-induced defects, strain, or the existence of charge defects.

## Phonon-mediated dephasing

The EDPHS radiation interacting with the flake induces a coherent polarization in the flake, due to the generation of a coherent superposition of quantum states. This aspect is similar to a $\pi/2$ pulse used in spin-echo experiments[54], which creates a coherent superposition between ground and excited states. The moving electron further interacts with the flake with a given time delay with respect to the EDPHS pulse, where the induced polarization stimulates the electron to produce coherent CL radiation. Thus, in contrast to spin-echo measurements and multi-dimensional spectroscopy schemes[55], which are based on multiple excitation schemes and highly nonlinear processes (four-wave mixing), our technique here relies on the different mechanisms of radiation from electron beams interacting with the sample to generate coherent and incoherent CL. In this way, the already generated coherent CL further interferes with the EDPHS polarization in the sample, resulting in prominent interference fringes within the energy-momentum CL map (Fig. 3a).

The coherent superposition generated by the EDPHS radiation decays over time within the dephasing time scale of the induced phonon polarization. Therefore, the generated CL signal has only a coherent nature within the time scale in which the EDPHS-induced-polarization maintains its coherence. Thanks to the remarkable mutual coherence between the EDPHS light and the near-field distribution of the moving electron, a high visibility of the order of $F(\tau = 0) = 0.57$ for the interference fringes is observed, where $\tau = 0$ is associated with the time within which both electron beam and the peak of the EDPHS radiation reach the sample at the same time. This is possible due to the retardation effect in the EDPHS structure and the time frame in which the induced polarization in the EDPHS contributes to the radiation[34].

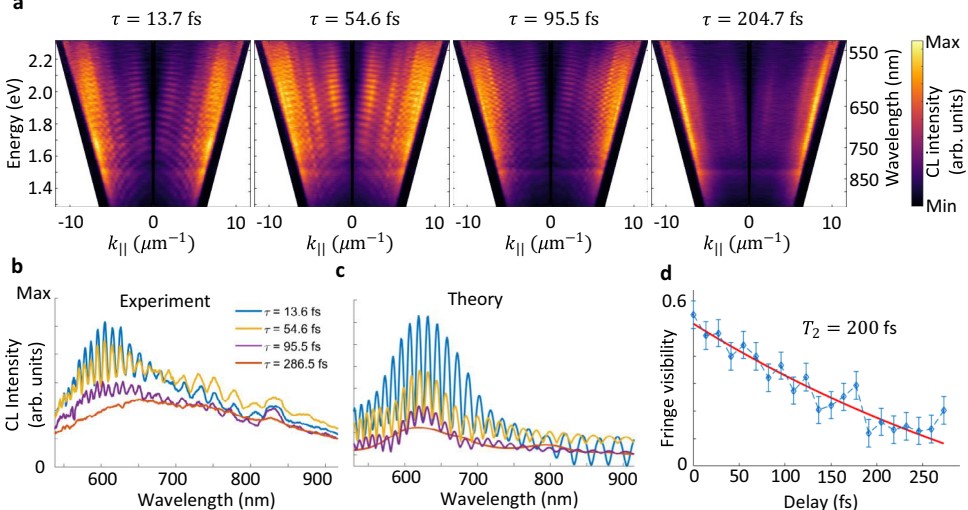

**Fig. 3 | Momentum-resolved CL intensity maps. a** Measured momentum-resolved CL intensity maps at depicted delays. Here, $k_{\parallel} = k_0 \sin\theta = \sqrt{k_x^2 + k_y^2}$ is the parallel wave number and $\theta$ is the polar emission angle with respect to the normal to the sample plane. **b** Measured CL intensity spectra at different delays for $\theta = 10° \pm 2°$. **c** Calculated polarization at the corresponding delays, indicating the vanishing of coherence at delays significantly longer than the dephasing time. **d** Plot of the fringe visibility versus delay. An exponential fit (red line) to the data reveals the dephasing time $T_2 = 220$ fs. The CL signals are taken from the positions marked in Fig. 2b by the purple box. Source data are provided as a Source Data file.

The visibility of the interference fringes, measured as

$$F(\tau) = (I_{\max}(\tau) - I_{\min}(\tau))/(I_{\max}(\tau) + I_{\min}(\tau)) \qquad (3)$$

gradually decreases with the time delay $\tau$ between the electron beam and the EDPHS radiation. Here, $I_{\max}$ and $I_{\min}$ are the maximum and minimum intensities of the CL signal at a given time delay. The observed interference fringes are most prominent at the wavelength associated with the phonon sidebands. We control the delay with the piezo stage in steps of 12 fs, which allows us to examine the interference fringes with sufficient time resolution. The line profiles of the spectral interferences at specific delays and within the angular range of $10° \pm 2°$ better indicate the fading of the visibility of the interference fringes over time (Fig. 3b), which also agrees well with the theoretical model based on the generation of a coherent CL signal due to the interaction with the EDPHS-induced coherent phonon polarization (Fig. 3c). Furthermore, the visibility of the interference fringes versus the delay $\tau$ shows an exponential decay, allowing us to measure the dephasing time of $T_2 = 200$ fs for the phonon polarizations (Fig. 3d). It is important to notice that the spectral interference fringes do not change their spectral period upon different delay times. This rules out that our observed fringes stem simply from quantum beats, which would result in their spectral frequency being inversely proportional to $\tau$. To better estimate the consistency in determining the coherent and incoherent CL signals, an inverse Fourier transform along the photon energy axis was used and the ratio of the broadening of the AC term to the peak time was calculated to estimate the error bar as shown in Fig. 3d.

Remarkably, angle-resolved spectral maps (see Fig. 3a) show the angular ranges into which different excitations in the sample emit photons. Phonon transitions in the wavelength range of 550 nm to 680 nm emit in the angular range of $\theta = 8°$ to $\theta = 16°$, while the electronic transition peaking at the wavelength of 880 nm emits most significantly in higher angular ranges $\theta > 60°$ (outer edges of the cone), further confirming the excitation of an electric dipole moment perpendicular to the surface of the flake.

## Population decay

In contrast to the dephasing dynamics of the coherent CL signal, the decay of the incoherent CL signal is related to the decay of the population $T_1$. This is due to the fact that the intensity of the incoherent CL is directly related to the generated population in the system, which further decays and releases CL signal. EDPHS radiation interacting with the sample increases the carrier density in the excited states, leading to an increase in the intensity of the CL signal compared to pure electron beam or EDPHS excitation. As the generated EDPHS-induced population decays over time, the CL intensity drops to an incoherent summation of the EDPHS spectrum and the CL spectrum from the sample after a long delay between the EDPHS and sample excitation.

To uncover the population decay $T_1$, the delay between the EDPHS and the electron beam arriving at the sample was varied at the steps of only 120 as, by measuring the integrated CL spectrum over the entire angular range of the emission above the sample, with a collection efficiency of $1.46 \pi$ sr (Fig. 4a, top). The CL intensity shows an exponential decay, in good agreement with theoretical calculations based on the expectation value of the number operator (Fig. 4a, bottom), which corresponds to the CL intensity when the EDPHS radiation is included in the interaction Hamiltonian.

The population relaxations for the different transitions observed are slightly different. While the first phonon state peaking at 805 nm shows an ultrafast population decay of only 289 fs, the decay corresponding to the second phonon state is significantly longer (Fig. 4b). Theoretically, the contribution to dephasing from population relaxation is $T_1/2$, which is 292 fs for the phonon transitions. This is slightly larger than the value of 200 fs for the measured dephasing time, due to the phonon-phonon coupling and rephasing processes occurring in the ensemble of phonon states.

Since the phonon states are energetically tightly packed in the wavelength range from 520 nm to 700 nm, a significant broadening of the CL signal is observed, due to free-induction decay and weak coupling between the phonon energy states in this range. In addition, the quantum-path interferences in the EDPHS-induced and electron-induced excitation and decay paths lead to significant spectral fluctuations in this region (Fig. 4c).

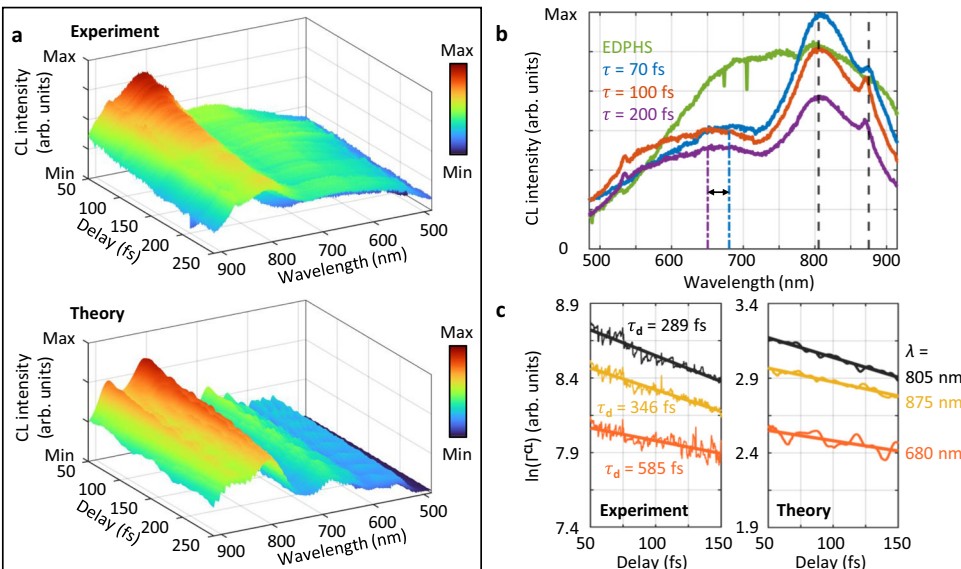

**Fig. 4 | Experimental and theoretical wavelength-delay maps of the CL response of the sample for EDPHS and electron beam excitations. a** (Top) Experimental and (Bottom) theoretical CL intensity spectra versus the delay $\tau$ between the EDPHS radiation and the electron arriving at the sample. **b** Measured CL intensity integrated over the entire angular range at depicted delays. The EDPHS radiation is indicated by the green line. The CL signals are taken from the positions marked in the inset of Fig. 2a. **c** Logarithm of the (Left) measured and (Right) calculated CL intensity ($\Gamma^{CL}$) of three main emission wavelengths versus the delay $\tau$. An exponential function (solid lines) is fitted to the data to obtain the corresponding damping times $\tau_d$ for each wavelength. Source data are provided as a Source Data file.

## Discussions

Exploring and controlling the dephasing time of single quantum emitters, implemented in solid-state systems and coupled to integrated photonic devices, is a key aspect in the further development of quantum technology and computation. Generally, revealing the dephasing dynamics in optical systems requires highly nonlinear processes, including four-wave mixing and techniques such as multi-dimensional electronic spectroscopy[56].

The decoherence process manifests itself in the interferometry techniques as a degradation of the visibility of the interference fringes[57]. This provides a powerful technique, for example using two-photon interference[58], to map the dephasing time of quantum systems.

However, in all these techniques, coupling to single defects has been proven to be challenging. Our method here, which is based on the interference effect in the sequential interaction of photons generated by the EDPHS and the sample, provides a significant improvement in addressing individual quantum systems and defects with high spatial and temporal resolution, as demonstrated here by applying it to defects in a thin hBN flake.

The dephasing and population decays allocated to our defects in the hBN flakes are significantly faster than the decay time reported by all-optical techniques. The population decay for the hBN defects reported so far was in the range of a few nanoseconds. This is mainly due to the fact that PL experiments are performed using either cw light or light pulses with a much narrower bandwidth compared to the EDPHS radiation, which precludes coupling to higher energy phonon states and superposition generation, as well as coupling to coherent phonons (See Supplementary Note 5 and Figs. 12, 13). Particularly, a major factor underlying the ultrafast dephasing time of the emitters observed here is due to the excitation of coherent phonon polaritons and their coupling to defects, with their propagation mechanisms and radiative nature leading to a faster decoherence mechanism for the emitter, as well as an enhanced population decay. In particular, electron beams, due to their ultrabroadband excitation mechanisms, provide an efficient way to simultaneously couple to both phonon polaritons in the far-infrared and electronic transitions in localized defects in the visible. It should also be noted that the emitters studied

here show a strong coupling to coherent phonon polaritons, in contrast to the emitter centers emitting at shorter wavelengths (see Supplementary Note 4 and Supplementary Fig. 11). In particular, for the latter class of defects, the emission does not show a coherent nature, mainly due to their weak coupling to coherent phonon excitations, which precludes the possibility of studying their dephasing dynamics with the multi-sequential CL technique proposed here.

Our results here have been based on the excitation of single emitters with electron beams, agreeing well with our theoretical framework that emphasizes coupling to single emitters. The focus remained on the exploration of the relaxation dynamics of single emitters. However, the system and methods developed can be applied to exploration of the dynamics of multiple emitters, when the density of the emitters increases (See Supplementary Note 6 and Supplementary Figs. 14 and 15). While ensuring a long dephasing time for single emitters is important for quantum technologies based on interferometry techniques, coherent phonon polarization in the hBN offers a wealth of possibilities, to enable quantum-sensitive measurements based on novel types of correlations in matter. Coherent phonons lead to an enhanced coupling between emitters, enabling emergent synchronization phenomena. For this to happen, one could consider coupling the emitters to photonic cavities with their resonant modes taking place within the upper Reststrahlen band (See Supplementary Note 5). They lead to novel types of superradiance in hBN flakes with a high density of emitters but need to be further investigated.

Our method thus allows the exploration of a rich set of physical phenomena, from single-emitter dephasing of quantum emitters in general to phonon- and photon-mediated correlations, including polaritons in different van der Waals materials, correlations in hybrid two-dimensional materials, and Moiré-induced polaritons and nonlinearities. This could pave the way to a better understanding of the emerging phenomena and localization effects in deep-subwavelength systems, but also in systems at mesoscopic scales. The experiments conducted here provide the first proof-of-concept demonstrating the applicability of our sequential CL technique for investigating the detailed dynamics of a single quantum emitter. However, further

improvements in the technical design could enhance accessibility to various emitters, such as by incorporating a scanning piezo stage for the sample holder. In other words, the scanning functionality of the SEM cannot be utilized in our current setup, as it would simultaneously scan the EDPHS structure.

Our technique transcends the methods available to explore dephasing dynamics by incorporating both luminescence spectroscopy and interferometry in a single scheme. The all-optical analog of this method, which includes photoluminescence (PL) spectroscopy and Mach-Zehnder interferometry, also allows for the exploration of ultrafast dephasing dynamics[59]. However, the sequential CL spectroscopy reported here offers greater flexibility in scanning materials at 1 nm spatial resolution and accessing randomly positioned defects in two-dimensional materials. Moreover, leveraging both coherent and incoherent interactions of electron beams with defects, we are able to recover not only dephasing time, but also population decay in a single experiment.

The interaction of the EDPHS radiation with the sample is similar to the incorporation of coherent radiation in Ramsey-type interferometry schemes preparing the sample in a coherent superposition[60]. The temporal duration within which the system freely evolves allows for altering the relative phase between the components of the superposition. In contrast with Ramsey-type interferometry though, our second pulse incorporates an electron beam, allowing for both coherent and incoherent interactions, where both the dephasing time and population decay are retrieved.

## Methods
### Liquid exfoliation
Hexagonal boron nitride (hBN) crystals were purchased from the HQ Graphene Company. To produce thin nanosheets, a liquid phase exfoliation process was applied to bulk hBN in isopropanol (Merck, ≥99.8%). The exfoliation process used an ultrasonicator (320W, Bandelin Sonorex, RK100H) with a timer and heat controller to prevent solvent evaporation. Sonication was performed in an ice bath using a cycle program of 5 min on followed by 1 min off, for a total of 180 min. The resulting suspension was drop-casted onto a holey carbon mesh grid for subsequent characterization.

### Cathodoluminescence imaging
All measurements detailed in this investigation involving cathodoluminescence (CL) spectroscopy, angle-resolved, and energy-momentum techniques were conducted utilizing the ZEISS Sigma field-emission scanning electron microscope with an attached Delmic SPARC CL system.

Throughout the entire experimental process, the electron microscope was consistently operated at an acceleration voltage of 30 kV unless otherwise specified. Two specimens were utilized simultaneously for phase-locked photon-electron spectroscopy. The upper one was an EDPHS-producing collimated light maintained by a nano-positioning system from SmarAct GmbH with three axial degrees of freedom, 1 nm step size accuracy, and a dynamic range of 12 mm. The lateral and vertical positions were precisely controlled with respect to the sample (See Supplementary Note 2 and Supplementary Fig. 4).

The second sample was a thin layer of hBN crystal that was placed on a carbon TEM grid held by the SEM stage. The beam current was set to 11 nA during the measurements. The CL detector was an aluminum parabolic mirror positioned below both samples. This mirror efficiently collects the generated CL radiation and projects it onto a CCD camera. Its specifications include an acceptance angle of $1.46\pi$ sr and a focal length of 0.5 mm. For spectral selection of the CL light, bandpass filters can be inserted into the optical path. During the measurements, the acquisition time for each pixel was set to 250 ms for hyperspectral imaging and 30 s for angle-resolved imaging.

### TEM measurements
HRTEM and EELS measurements were performed in a JEOL ARM200CF transmission electron microscope equipped with a $C_s$ corrector in the imaging system. The TEM was operated at 200 kV. EELS spectra were recorded with a CCD camera attached to a Gatan Imaging filter (GIF Quantum ERS). Spectral imaging was performed in the scanning mode with an electron-probe size smaller than 0.5 nm. Spectral imaging is achieved by acquiring EELS data from each pixel within a 2D area and then extracting element-specific absorption edges. HRTEM images were acquired in parallel beam mode using a Gatan OneView camera. All data displayed are raw data.

## Data availability
The experimental raw data substantiating Fig. 1, Fig. 2b, Fig. 3, and Fig. 4 that support the findings of this study are available in "Zonodo" with the identifier https://doi.org/10.5281/zenodo.14832120[61]. Source Data for Figs. 1a, b, 2b, 3a, c, 4a are provided with this paper. Source data are provided with this paper.

## Code availability
The numerical code used to simulate the data is available from the corresponding author on request.

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

## Acknowledgements

N.T. and H.G. acknowledge fruitful discussions with J. Wrachtrup (Stuttgart University). N.T. acknowledges as well fruitful discussions with M. Kociak (CNRS, France). This project has received funding from the European Research Council (ERC) under the European Union's Horizon 2020 research and innovation program under grant agreement no.

802130 (Kiel, NanoBeam) and grant agreement no. 101017720 (EBEAM), and from Deutsche Forschungsgemeinschaft under Grant agreement nos. 525347396 and 447330010, and from Volkswagen Stiftung (Momentum Grant). M.H. and H.G. thank DFG, BMBF, and ERC grant (COMPLEXPLAS) for funding.

## Author contributions

M.T., P. B., and M.B. Performed the experiments and analyzed the data together with N.T. M.H. fabricated the EDPHS structure. N.T. conceived the data, performed the simulations, and wrote the manuscript with contributions from all coauthors. H.G., P.v.A., and C.K. contributed to the discussions. W.S. and B.H. performed the transmission electron microscopy imaging and electron energy-loss spectroscopy.

## Funding

## Competing interests

The authors declare no competing interests.
