## [Transparent Peer Review file · Nature Communications]

Ultrafast phonon-mediated dephasing of color centers in hexagonal boron nitride probed by electron beams

Corresponding Author: Professor Nahid Talebi

Version 0:

Reviewer comments:

Reviewer #1

(Remarks to the Author)

The manuscript reports an ultrafast dephasing time of color centers in hBN using phase-locked photon-electron spectroscopy technique, and attributes the reason to efficient excitation of coherent phonon-polariton by electron beam. The theory is interesting as understanding the behavior of defects in hBN under electron beam excitation is important for high-resolution CL spectroscopy. However, the connection between experimental data and theory should be elaborated with more details for publication in Nature Communications.

1. The authors made a good agreement of their theory with the experimental data using the 8-level quantum system. It would be great to discuss why it should be this number, what happens with the other number of levels, or if these phonon states also agree well with the proposed defect structure in hBN.

2. In which sample or position did the authors obtain data for Fig. 3 and 4? If it is a defect-related process, did the authors find a different decoherence time at another position without a defect, or at a different defect site?

3. Would the dephasing time increase if the authors make EDPHS of narrower bandwidth?

4. In Fig. 2a,b, the authors found three peaks and attribute the lowest energy peak to localized defects and the other two peaks to phonon-polariton coupling of these defects. However, the authors may need to provide additional discussion to support the coupling to phonon-polariton.

(1) In Fig 2b, CL excitation at the edge contains shorter wavelength peaks without defect-related 880 nm peak, then the former signal might be related to bulk hBN rather than defect coupling, or independent high-density defects in hBN?

(2) All three phonon peaks in Fig2c seem to exist only at the edges of the flake in Fig2d. Though E2 signal is narrower than the other two, it may be because of higher contrast, as there is stronger signal on the outside of hBN flake (upper-right region) than the thicker part of hBN (lower-left region) for E1 and E3 signal.

(3) The energy difference between three CL peaks are ~147 meV and ~295 meV, which are not in the range of either phonon peaks in EELS data (157-186 meV) or the upper reststrahlen band of hBN (169-200 meV).

5. The authors used an electron beam of up to 10nA and 30 keV, and there are several papers that suggest strong e-beam irradiation breaks the lattice of hBN and induces single-photon emission (Nature Communications 12, 3779 (2021), APL Mater. 11, 071108 (2023)). The authors may provide how it is different in their sample.

6. Minor typos:

(1) Line 209, scanning EELS image of the area marked by the orange box → scanning EELS image of the inset figure (There is clear feature that the image is not for the orange box but for the entire inset image)

(2) Line 210, fig (d) → fig (e)

(3) Line 182, fig 2d → fig 2b

Reviewer #2

(Remarks to the Author)

Taleb et al. report on the investigation of defect states in a thin layer of hexagonal boron nitride (hBN) using electron beam induced cathodoluminescence (CL) spectroscopy in a scanning electron microscope. They employ a double interaction scheme reported earlier in [Taleb et al. Nat. Phys. 19, 869–876 (2023)], where they used a transformation lens approach to produce coherent and incoherent CL radiation in the first interaction plane that is focused onto a second interaction plane containing a sample that is investigated. The second interaction plane is subject to the CL radiation created by the electron passing the first plane and the very same electron that excites the hBN at a later time. The phase between the excitations can be set by changing the distance between the two interaction planes via a piezo.

The main results are the observation of dephasing and decay times of phonon-mediated electronic (defect) states in hBN at room temperature, which are excited with the CL radiation and subsequently probed by electron-generated radiation and the interference of this radiation with the CL. The obtained dephasing and decay times are shorter than previously reported times from all-optical (e.g., PL) measurements. The difference is explained by the coherent excitation of a superposition of phonon-mediated states with the broadband CL radiation.

The general scheme is interesting, but in the present implementation, it doesn't go well beyond a (very complicated, electron-driven) linear spectroscopy setup in a scanning electron microscope. Similar results can be obtained by using ultrashort (e.g., few-cycle) laser pulses (having a sufficient bandwidth comparable to the CL radiation) in an all-optical pump-probe scheme. The authors mention that all-optical methods are limited to measuring only with ns-temporal resolution; however, fs-linear spectroscopy is well possible. The reason I find the approach still appealing is the possibility to measure, in principle, the (sub-) nanometer spatial distribution of a single defect state. As the authors also point out, this is the big advantage of using a scanning electron microscope. However, as long as this advantage is not utilized, I do not recommend the manuscript for publication. The results are also very similar to those presented in [Taleb et al. Nat. Phys. 19, 869–876 (2023)] but on a different sample system and with additional TEM measurements.

Further comments:

- The mentioned attosecond resolution is not shown in the data presented in the manuscript. More specifically, only a sub-cycle phase resolution is possible, which only translates to attosecond precision. The authors should be more careful in their formulation.
- The method of CL generation and focusing seems to lead to a not perfectly collimated beam (see Supplement), which should also show chromatic aberrations within the bandwidth of the CL radiation. Therefore, the observed changes in emission and interference intensity as a function of plane distance could also stem from these aberrations.

Version 1:

Reviewer comments:

Reviewer #2

(Remarks to the Author)

I have carefully reviewed the authors' response and the updated manuscript files. They have addressed the reviewers' questions and clarified most of the concerns.

While the revised manuscript now specifies where the time-resolved CL measurements were taken and includes a second data point (which shows no effect) in the supplemental material, it remains unclear why the authors have not fully utilized the inherent capabilities of the scanning electron microscope, particularly the scanning. At the very least, I would expect measurements from multiple defect centers and clusters, as significant variations in population decay and dephasing times could exist. While this paper is interesting as a proof of concept, a more controlled study with greater statistical significance would be preferable.

The authors have also clarified the distinction between their method and optical spectroscopy techniques. Although their arguments are valid, the method seems most appropriate in the context of spatially resolved measurements, which would reveal the behavior of single or multiple defects. The additional measurement of a defect showing no interference underscores this point. In other words, how can the authors be certain they are investigating a single defect without spatial information?

I believe, with further refinement and a more comprehensive study, this approach could offer significant insights into defect dynamics and might become a valuable tool for the field.

Reviewer #3

(Remarks to the Author)

The only question I have is to ask what is the time resolution (Instrument response function). For such a short time scales, relevant to dephasing the pulse duration of the pump and the probe must be below 100 fs. I haven't found this information.

Version 2:

Reviewer comments:

Reviewer #2

(Remarks to the Author)

I thank the authors for their very clear response and have no further questions or comments.

Reviewer #3

(Remarks to the Author)

I am pleased with how my comment has been addressed. The additional information I requested regarding the pulse duration of the excitation and probe beams, which determines the time resolution as been incorporated into the manuscript.

Response Letter

Reviewer 1:

The manuscript reports an ultrafast dephasing time of color centers in hBN using phase-locked photon-electron spectroscopy technique, and attributes the reason to efficient excitation of coherent phonon-polariton by electron beam. The theory is interesting as understanding the behavior of defects in hBN under electron beam excitation is important for high-resolution CL spectroscopy. However, the connection between experimental data and theory should be elaborated with more details for publication in Nature Communications.

Our response:

We thank the referee for her/his comments and the positive evaluation of our manuscript. We particularly appreciate the reviewer's statement regarding our theory and finding it an important step to clarify the excitation of defects by electron beams. We have now modified the text according to the constructive comments raised by the reviewer and believe that the manuscript now answers the reviewer's open questions.

Comment#1:

The authors made a good agreement of their theory with the experimental data using the 8-level quantum system. It would be great to discuss why it should be this number, what happens with the other number of levels, or if these phonon states also agree well with the proposed defect structure in hBN.

Our response:

We thank the reviewer for this constructive feedback. To determine the number of states, we have carefully compared the spectra predicted by theory for different numbers of states, ranging from 3 to 8, with both the experimentally obtained CL spectra and the combination of CL and EDPHS results. Particularly, the shape of the CL spectra is most accurately reproduced by choosing 8 states as stated in the manuscript. Moreover, the fast decay and the dephasing times occurring in this specific defect geometry are accurately described by this model. It should be noted that although it is possible to reconstruct the pure experimental CL spectrum of the defect with lower state numbers, this model could not reproduce the fast decay times we observed. Therefore, in order to better configure the chosen parameters for the theoretical model, we have carefully compared the theoretical results with the full set of experimental observations. We have now added more details to the Supplement to describe all these aspects.

We would like, however, emphasize here that we are currently preparing a manuscript that explains the theoretical approach and adapt it to various multilevel systems in full details, with the help of our Postdoc Dr. Hebrew Chrispin. He uses another approach and compares it with the platform we used here. This manuscript will be submitted soon to Physical Review Letters.

Actions taken:

We have added the following text and figures to the supplementary information:

“The states associated with phonon resonances are not energetically equidistant, which is explained by the anharmonic potential constraining molecular vibrations (described by the polynomial expansion or Morse potential^{5,6}). The eigenenergies of an anharmonic oscillator are generally expressed as $E(n) = -E_0 + \hbar\omega_0 \left(n + \frac{1}{2} \right) \left\{ 1 - \xi \left(n + \frac{1}{2} \right) \right\}$, where ξ and E_0 are parameters controlling the anharmonicity of the potential and ω_0 is the oscillation frequency of the oscillator. We first find that by choosing $E_0 = 0.090 \text{ eV}$, $\hbar\omega_0 = 0.119 \text{ eV}$, and $\xi = 0.012$, the obtained energies closely match our theoretically predicted eigenstates for an 8-level system (Fig. S3a). Moreover, the minimum potential energy of the harmonic oscillator associated with the excited electronic state with respect to the ground electronic state is calculated to be 1.3528 eV. These eigenstates are used in the theoretical model to reproduce three different sets of experimental data: CL spectra, incoherent CL spectra confirming the population decay, and coherent CL spectra obtained by angle-resolved CL spectroscopy.

Figure S3: Quantum states and calculated CL spectra associated with an N-level quantum system. (a) The potential and eigenenergies of an anharmonic oscillator. See text for details. Incoherent CL intensity versus wavelength and delay between the EDPHS and electron excitation for (b) a maximum number of 3 states, and (c) a maximum number of 6 states. (d) CL intensity associated with the induced polarization versus wavelength at the shown delays between the EDPHS radiation and the electron-beam excitation.”

To justify the choice of 8 states, we present the spectra of incoherent CL excitation (Equation (S.8)) for different cases where $N = 3$ states and $N = 6$ states are considered (Figure S3). It is evident that the inhomogeneous broadening associated with higher energy states (see Figure S1 and main text) is only accurately reproduced when a minimum of 6 states is considered. Moreover, the induced polarization captures the interference phenomena observed in the CL momentum-resolved spectra as a function of the delay between EDPHS radiation and electron-beam excitation (Fig. S3d) when 6 states are considered. However, both the spectral shape and the dephasing time are more accurately reproduced when 8 states are considered. For a system with only 6 quantum states, the dephasing time is significantly longer (Fig. S3d).”

The following references have been added to the supplementary as well:

- “5 Morse, P. M. Diatomic Molecules According to the Wave Mechanics. II. Vibrational Levels. *Physical Review* **34**, 57-64 (1929). <https://doi.org:10.1103/PhysRev.34.57>
- 6 Fues, E. Das Eigenschwingungsspektrum zweiatomiger Moleküle in der Undulationsmechanik. *Annalen der Physik* **385**, 367-396 (1926). <https://doi.org:https://doi.org/10.1002/andp.19263851204>”

Comment#2:

In which sample or position did the authors obtain data for Fig. 3 and 4? If it is a defect-related process, did the authors find a different decoherence time at another position without a defect, or at a different defect site?

Our response:

We thank the reviewer for this comment. We have performed several experiments to identify the nature of the particular defect we are studying and to confirm that the processes we observe here are related to the defects and their coupling to phonons. In fact, to observe interference patterns, we need to have coherent emission from the sample. A single defect coupling to electron beams does not produce light in a coherent quantum state. In particular, the expectation value of the electric field would be zero, and therefore no first-order interferences are expected. In the case of the defects studied here, the coherent excitation of phonon states by both the EDPHS radiation *and* their strong coupling to the defects generate a coherent polarization in the sample. To better clarify this aspect, but also better explain the role of coherent phonon excitations in the fast dephasing time, we have performed similar experiments on parts of the sample where the defects at the longer wavelength of 880 nm are excited at a much lower rate compared to the defects that we observe at the wavelength of 530 nm. The latter defects do not show strong coupling to coherent phonon polaritons. Most remarkably, the interference effects within the short time scales that we observe for the longer-wavelength defect disappear completely from the acquired CL data. We have now added these experiments as well to the Supplement to better explain both the uniqueness of the dephasing dynamics we observe for the specific defects around 530 nm we study here, and to emphasize the role of coherent phonon excitations in the formation of the interference fringes.

Actions taken:

We have added the following sentence to the captions of Figures 3 and 4 to indicate the position where we acquired the CL signals:

“The CL signals are taken from the positions marked in Figure 2b by the purple box.”

We added the following text and figure to the Supplement:

“Supplementary Note 4. Dephasing dynamics of the defects emitting at the wavelength of 530 nm

The peculiar ultrafast dephasing nature of the defect emitting at 880 nm is related to its strong coupling to coherent phonon polariton excitations in the sample. Notably, we also observe a second class of defects in our liquid-exfoliated hBN flakes that emit at 530 nm (zero-phonon line; see Fig. S10b). This emission is accompanied by two phonon lines at 553 nm, and 566 nm, indicating an energy difference of 97.3 meV and 158.9 meV between the zero-phonon line and the subsequent phonon lines, respectively. This particular defect has been extensively studied in the literature (see references [8] to [20] in the main text) and is known to have a long decay time of a few nanoseconds, in contrast to the defects we have studied so far.

Figure S10: Investigation of the dephasing time of the defect emitting at $\lambda = 530 \text{ nm}$. (a) The secondary electron dark-field image of the EDPHS and an hBN flake below it, positioned 20 μm away from the EDPHS. The electron beam is focused on the EDPHS. (b) The CL spectra of the EDPHS (green shaded area) and the combination of EDPHS and sample integrated over the shaded box in the panel (red shaded area) (a). (c) Hyperspectral CL images of the combination of the EDPHS and hBN flake at the wavelength shown. (d) Momentum-resolved CL spectral maps at the indicated delays between the EDPHS and the sample radiation.”

To study this second class of defects in our hBN flakes, we utilize an EDPHS structure with a hole in the center that sustains a broadband spectrum as shown in Fig. S4c (see green part of Fig. S10b). The CL spectrum of the combination of the hBN and EDPHS is also shown in Fig. S10b in red, demonstrating efficient excitation of the emitting center at 530 nm. The spectral profile of the EDPHS structure is perfectly suited for the excitation of the coherent phonon lines at 797 nm and 670 nm. In addition, a faint spectral feature associated with the defect centers emitting at 880 nm is observed, shifted to 860 nm due to the high-intensity phonon lines covering the emitter wavelengths.

Hyperspectral images associated with $\lambda_1 = 630$ nm, $\lambda_2 = 670$ nm, $\lambda_3 = 797$ nm, $\lambda_4 = 860$ nm indicate that a single defect emitting at $\lambda_4 = 860$ nm is effectively excited by the combination of the EDPHS and the hBN flake. However, the CL intensity associated with the emission at $\lambda_1 = 630$ nm is more uniformly distributed within the sample, increasing the efficiency of coupling to this emitter at this sample location.

This emitter is not coupled to coherent phonon lines at $\lambda_2 = 670$ nm and $\lambda_3 = 797$ nm. This is more evident from the momentum-resolved CL spectral maps. The interference fringes observed in the experiments reported in Figure 3 of the main text are absent from the acquired CL energy-momentum maps shown in Figure S10d, highlighting the lack of a coherent emission mechanism from these defects.”

We have also added the following text to the main text:

“It should also be noted that the emitters studied here show a strong coupling to coherent phonon polaritons, in contrast to the emitter centers emitting at shorter wavelengths (see Supplementary Note S4). In particular, for the latter class of defects, the emission does not show a coherent nature, mainly due to their weak coupling to coherent phonon excitations, which precludes the possibility of studying their dephasing dynamics with the multi-sequential CL technique proposed here.”

Comment#3:

Would the dephasing time increase if the authors make EDPHS of narrower bandwidth?

Our response:

We thank the reviewer for this insightful comment. Indeed, we expect that the incorporation of a narrow-line EDPHS would allow the selective excitation of different state superpositions and a systematic and careful analysis of the dephasing dynamics. The EDPHS structures we have fabricated so far produce a rather intense and collimated light beam. The emission characteristics of the EDPHS are a compromise between the control of the spatial profile and the light intensity. As a result of the ensemble of holes (their sizes, spacings, and positions), a variety of photonic states are observed, leading to broadband and chirped excitation.

To control the bandwidth of the emission, we would have to completely change the design principles and explore three-dimensional geometries based on the phase-matching principle between the electron beams and the sample excitations, or use hybrid material systems that rely on the combination of specific material excitations, such as excitons coupled to plasmons. Both of these approaches are ongoing research directions in our group and will be the subject of another manuscript that we are preparing.

**Comment#4:**

In Fig. 2a,b, the authors found three peaks and attribute the lowest energy peak to localized defects and the other two peaks to phonon-polariton coupling of these defects. However, the authors may need to provide additional discussion to support the coupling to phonon-polariton.

(1) In Fig 2b, CL excitation at the edge contains shorter wavelength peaks without defect-related 880 nm peak, then the former signal might be related to bulk hBN rather than defect coupling, or independent high-density defects in hBN?

Our response:

We thank the reviewer for this comment. First, hBN is a transparent material in the visible range with a refractive index that changes from 2.183 to 2.086 within the wavelength range of 450 nm to 1000 nm. Thin films of this material cannot support guided modes. For bulk excitation, the material should either support bulk material excitations such as volume plasmons, excitons, band-gap transitions, or support excitation of Cherenkov radiation. hBN cannot support any of these excitations in the visible wavelength range. The refractive index of this material is even lower than the Cherenkov threshold, namely $\cos(\theta) = n/\beta$, where θ is the angle of Cherenkov radiation with respect to the propagation direction of the electron beam, n is the refractive index of the material, and $\beta = v_e/c$, where v_e is the group velocity of the electron, and c is the speed of light. For an electron traveling at the kinetic energy of 30 keV, the refractive index should be larger than 3.045 for Cherenkov radiation to occur. Therefore, we exclude the possibility of having any bulk excitation, which is in perfect agreement with our simulations (not shown here, since no CL or electron-energy loss peak is observed).

Second, direct electronic transitions in the defects are coupled to phonon polaritons, leading to the excitation of the latter. Like any other in-plane polaritons, the emission of phonon polaritons from the edges happens to be more efficient, resulting in the observed high-intensity CL signal from the edges. Therefore, the high-intensity signal occurring at the edge, even though the defects are not directly positioned at the edge, is related to the excitation of polaritons by defects and their in-plane propagation and scattering from the edges.

Comment#5:

(2) All three phonon peaks in Fig2c seem to exist only at the edges of the flake in Fig2d. Though E2 signal is narrower than the other two, it may be because of higher contrast, as there is stronger signal on the outside of hBN flake (upper-right region) than the thicker part of hBN (lower-left region) for E1 and E3 signal.

Our response:

We thank the reviewer for this insightful comment and for allowing us to better explain the excitation of phonon polaritons in our hBN flakes. First, we would like to mention that the E2 peak is observed not only in thinner regions of the flake, but also in vacuum at an impact position of a few nanometers away from

the edge of the flake (see Figures S11 and S12 below). Such behavior can only be attributed to near-field excitations, where the given frequency range, can be attributed to phonon polaritons. In the following, we provide evidence for the excitation of hBN phonon polaritons and compare them with theoretical results.

Actions Taken:

To better explain the excitation of phonon-polaritons in the sample, we have added the following section to the Supplement:

“Supplementary Note 5. Phonons and phonon-polaritons in hBN flakes

Finally, we give an overview of phonon-polaritons excited in our hBN flakes. Within two frequency ranges in the Reststrahlen bands, hBN hosts hyperbolic phonon polaritons. The permittivity of the material along the in-plane and out-of-plane directions has opposite signs in the regions highlighted in Fig. S11 a, leading to different type I and II hyperbolic responses for the regions within $\hbar\omega_{\text{TO},\perp} = 0.097\text{eV}$ and $\hbar\omega_{\text{LO},\perp} = 0.103\text{eV}$ (shaded in green), and $\hbar\omega_{\text{TO},\parallel} = 0.17\text{eV}$ and $\hbar\omega_{\text{LO},\parallel} = 0.20\text{eV}$ (shaded in red), respectively¹¹. The former and the latter are attributed to the lower and upper Reststrahlen bands, respectively. Given the growth conditions of the flakes, the in-plane and out-of-plane directions correspond to the parallel and the normal directions, respectively, relative to the surface of the flakes.

The electron beam can excite bulk LO and TO excitations in the lower Reststrahlen band as it traverses the hBN flake in the direction normal to the flake (see Fig. S11b), resulting in two electron energy-loss peaks at the corresponding energies. However, hBN films cannot host any guided waves or surface phonon polaritons in the lower Reststrahlen band. Notably, the energy shift of the first phonon line corresponding to the lower wavelength emitters is in good agreement with $\hbar\omega_{\text{TO},\perp} = 0.097\text{eV}$. We conclude that these classes of emitters efficiently couple to phonons in the lower Reststrahlen band.

In contrast, in the upper Reststrahlen band, hBN films as thin as 10 nm can host hyperbolic phonon polaritons. The dispersion of these polariton waves is captured in the calculated momentum-resolved electron energy-loss spectra (Fig. S11c). In addition to phonon polaritons, a distinct signature is observed at $\hbar\omega_{\text{LO},\parallel} = 0.20\text{eV}$, corresponding to the excitation of bulk LO phonons. Finally, when considering bulk excitations (boundary effects are neglected), a clear signal is observed in the calculated momentum-resolved electron energy-loss spectra, corresponding to the Cherenkov radiation (Fig. S11d).

The lower Reststrahlen band is hardly accessible due to the broadening of the zero-loss peak, when performing low-energy electron energy-loss spectroscopy. However, as shown in the main text, phonon-polariton excitations in the upper Reststrahlen band is clearly accessible with this technique. To better determine the excitation of both phonon-polaritons and bulk LO phonons, low-energy electron energy-loss spectra are now acquired in an aloof excitation, where bulk phonons are not excited, and the results are compared to the case where the electron beam traverses the flake (Figure S12). Even in vacuum and at a distance of 10 nm from the flake, electron beams can excite surface phonon polaritons, leading to a sharp peak at the energy of $E_2 = 0.186\text{eV}$ corresponding to surface phonon polariton excitation, where the strongest signal is also observed in simulations (compare Fig. S12a with Fig. S11c). Bulk LO phonons also appear in the EELS signal when the electron beam traverses the material at a given thickness (Fig. S12 b and c). A lower energy excitation is also observed at 0.157 meV, which is below the upper Reststrahlen band and corresponds to Cherenkov radiation.

The energy of the first phonon line, corresponding to the emitters emitting at 880 nm, exhibits an energy shift of about 0.147 eV, with respect to the zero-phonon line. However, this peak has a wide bandwidth, so that the energy shift with respect to the zero-phonon line ranges from 0.11 eV to 0.22 eV. This energy shift is significantly larger than $\hbar\omega_{\text{TO},\perp} = 0.097\text{ eV}$ and more pronounced than the upper Reststrahlen band. However, due to the clear signals observed in the low-energy electron energy-loss spectra corresponding to the Cherenkov radiation, we would not rule out the influence of the Cherenkov radiation in the observed coherent interactions.

Figure S12: Phonon-polaritons in hBN. (a) Permittivity of hBN, demonstrating the anisotropic nature of the materials and its hyperbolic behavior within the two Reststrahlen bands. The real and imaginary parts of the permittivity are indicated by solid and dashed lines, respectively. The atomic structure of the material is shown in the inset. The in-plane and out-of-plane permittivity components are shown as blue and red lines, respectively. (b - d) Calculated momentum-resolved electron energy-loss spectra in the (b) lower and (c) upper Reststrahlen bands and in a bulk hBN medium without considering the boundary effects. The electron has a kinetic energy of 30 keV and traverses an hBN thin film in the direction normal to the surface of the flake. The thickness of the flake is 10 nm.

Figure S12: Position dependence of the low-energy electron energy-loss spectra. The upper figures are the dark-field TEM images and the lower figures depict the low-energy electron energy-loss spectra at the positions marked by the orange box. (a) Aloof excitation, where only a single peak at $E = 0.186$ eV is observed, corresponding to surface-phonon polaritons scattered from the edge of the flake. (b, c) As electrons traverse the flake, bulk excitations are also excited, resulting in peaks corresponding to LO phonons and also Cerenkov radiation.”

The following reference has also been added to the Supplement:

“[11] Boll, M. K., Radko, I. P., Huck, A. & Andersen, U. L. Photophysics of quantum emitters in hexagonal boron-nitride nano-flakes. *Opt. Express* **28**, 7475-7487 (2020).
<https://doi.org/10.1364/OE.386629>”

Comment#6:

(3) The energy difference between three CL peaks are ~ 147 meV and ~ 295 meV, which are not in the range of either phonon peaks in EELS data (157-186 meV) or the upper reststrahlen band of hBN (169-200 meV).

Our response:

We thank the reviewer for this comment. Although the position of the CL peaks exhibits an energy shift corresponding to the values given by the reviewer, these peaks are significantly broadband and would cover the range of 0.11 eV to 0.22 eV for the first peak. This energy range covers the upper Reststrahlen band of hBN as explained in our response to the comment above. Note that the second peak appears as a replica of the first peak due to the quantum localization effects formed by the phonons trapped in the (an)harmonic potential of the defects.

Actions Taken:

We have added the following paragraph to the Supplementary Information:

“The energy of the first phonon line, corresponding to the emitters emitting at 880 nm, exhibits an energy shift of about 0.147 eV, with respect to the zero-phonon line. However, this peak has a wide bandwidth, so that the energy shift with respect to the zero-phonon line ranges from 0.11 eV to 0.22 eV. This energy shift is significantly larger than $\hbar\omega_{TO,\perp} = 0.097\text{eV}$ and more pronounced than the upper Reststrahlen band. However, due to the clear signals observed in the low-energy electron energy-loss spectra corresponding to the Cherenkov radiation, we would not rule out the influence of the Cherenkov radiation in the observed coherent interactions.”

Comment#7:

5. The authors used an electron beam of up to 10nA and 30 keV, and there are several papers that suggest strong e-beam irradiation breaks the lattice of hBN and induces single-photon emission (Nature Communications 12, 3779 (2021), APL Mater. 11, 071108 (2023)). The authors may provide how it is different in their sample.

Our response:

We thank the reviewer for this comment. It should be mentioned that our initial expectation was indeed that this peculiar defect that we observed in our SEM could be either due to electron-beam-induced strain or eventually due to sample damage. However, our SEM images of the sample taken before and after the CL acquisitions, as well as the photoluminescence spectroscopy performed before and after the CL acquisitions and their comparison with each other, rule out the influence of the electron-beam radiation damage to produce such defects.

Indeed, we can selectively create such defects that emit at completely different wavelengths by exposing the sample to intense electron beam irradiation for several seconds at the same sample position. However, we typically use an exposure time of a few ms and careful scanning to avoid damaging the sample. In addition, hBN has a more stable material configuration compared to other materials such as perovskites. For example, the image below shows how a perovskite sample is affected by electron beam acquisition, and also how we can avoid such an effect by encapsulating the perovskites with hBN flakes.

Figure 2: The effect of electron beam on Ruddlesden-Popper perovskite (RPP) flakes. (a) An electron-beam probed RPP flake that is severely damaged after scanning with an electron beam to obtain spectral images. (b) The SEM

image of RPP flakes encapsulated by a large hBN flake. (c) The image of the sample after several minutes of electron beam irradiation, where the the part of the sample encapsulated by hBN is still stable.

Comments about typos:

6. Minor typos:

(1) Line 209, scanning EELS image of the area marked by the orange box → scanning EELS image of the inset figure (There is clear feature that the image is not for the orange box but for the entire inset image)

Our response:

We thank the reviewer for this comment and apologize for the error. We have now corrected this sentence in the revised version.

Actions taken:

We have corrected this sentence:

“scanning EELS image of the area marked by the orange box...”

to

“Scanning EELS images of the *inset figure in panel (c)*”

(2) Line 210, fig (d) → fig (e)

Our response:

We thank the reviewer for this comment and apologize for the error. We have corrected the typo.

Actions taken:

We have corrected (d) to (e) in line 210.

(3) Line 182, fig 2d → fig 2b

Our response:

We thank the reviewer for this comment and apologize for the error. We have also corrected the typo.

Actions taken:

We have corrected Fig. 2d in line 182 to Fig. 2b.

**Reviewer #2****Comment#1:**

Taleb et al. report on the investigation of defect states in a thin layer of hexagonal boron nitride (hBN) using electron beam induced cathodoluminescence (CL) spectroscopy in a scanning electron microscope. They employ a double interaction scheme reported earlier in [Taleb et al. Nat. Phys. 19, 869–876 (2023)], where they used a transformation lens approach to produce coherent and incoherent CL radiation in the first interaction plane that is focused onto a second interaction plane containing a sample that is investigated. The second interaction plane is subject to the CL radiation created by the electron passing the first plane and the very same electron that excites the hBN at a later time. The phase between the excitations can be set by changing the distance between the two interaction planes via a piezo.

The main results are the observation of dephasing and decay times of phonon-mediated electronic (defect) states in hBN at room temperature, which are excited with the CL radiation and subsequently probed by electron-generated radiation and the interference of this radiation with the CL. The obtained dephasing and decay times are shorter than previously reported times from all-optical (e.g., PL) measurements. The difference is explained by the coherent excitation of a superposition of phonon-mediated states with the broadband CL radiation.

Our response:

We kindly thank the reviewer for highlighting some of the features of the technique we have developed and also his/her time in reviewing our manuscript. We would agree with the reviewer that the fast dephasing time of this peculiar defect that we study is a specific novelty of our manuscript. Besides this, we have also developed a theory for explaining coherent and incoherent CL from these defects, for the first time. Please notice that our technique is currently the only one that can study the dephasing dynamics of solid-state based excitations at such short time scales (1.4 fs temporal resolution) and at true nanoscale (1 nm spatial resolution).

Comment#2:

The general scheme is interesting, but in the present implementation, it doesn't go well beyond a (very complicated, electron-driven) linear spectroscopy setup in a scanning electron microscope. Similar results can be obtained by using ultrashort (e.g., few-cycle) laser pulses (having a sufficient bandwidth comparable to the CL radiation) in an all-optical pump-probe scheme. The authors mention that all-optical methods are limited to measuring only with ns-temporal resolution; however, fs-linear spectroscopy is well possible.

Our response:

We thank the reviewer for this comment and for finding our method interesting. However, we believe our technique surpasses being merely a linear spectroscopy technique.

First, if we compare our scheme with light-based spectroscopy schemes, we would need to perform photoluminescence (PL) spectroscopy, excite the sample with a sub-cycle high-intensity laser beam tuned to phonon lines, filter the emission at the wavelength of the excitation (since electron probe techniques do not produce any background emission), and pass the emission from the defects through a Mach-

Zehnder interferometer (Nature 419, 594 (2002)). Such a technique does not allow coupling to single emitters unless the sample contains only a few defects spaced a few micrometers apart. Moreover, although PL is typically a linear optical process, it involves multiple spontaneous and sequential relaxations. In particular, coupling through eight states and forming a quantum walk over these states require multi-photon processes. Our technique combines both setups into a single scheme: it is a combination of PL linear spectroscopy and interferometry.

Second, there is a significant difference between our method and optical linear spectroscopy. Our technique benefits from both coherent and incoherent cathodoluminescence. Incoherent cathodoluminescence, which has no equivalent in femtosecond linear spectroscopy, is related to the incoherent creation of population and is directly linked with the diagonal terms of the density matrix, as explained in the manuscript. In contrast, coherent cathodoluminescence is linked to the induced linear polarization in the sample. Therefore, by discriminating between coherent and incoherent cathodoluminescence, one can reveal both T_1 and T_2 times, which is not possible with all-optical linear spectroscopy techniques.

Our approach to experimentally differentiate coherent and incoherent cathodoluminescence is angle-resolved spectroscopy. The visibility of the interference fringes in the angle-resolved maps reveals the degree of mutual coherence between the EDPHS radiation (a coherent radiation) and the radiation from the defect. If a moving electron excites the defect alone, there is no coherent radiation. Therefore, interference fringes are not observed in the far-field angle-resolved map, as shown in the new Fig. S6.

EDPHS radiation brings the system into a coherent superposition of the excited states, similar to a $\pi/2$ pulse in optics. This approach is comparable to Ramsey-type experiments, where two sequential pulses interact with the system, and depending on the free-evolution temporal duration between the pulses, the phase difference between the superposition components is controlled. This is similar to the experiments performed by Serge Haroche and collaborators for retrieving the decoherence (Nature 455, 510 (2008)). This coherent superposition induces linear polarization inside the sample, and therefore, the CL signal also exhibits coherent radiation. The coherent part of the CL radiation can then interfere with the EDPHS light, resulting in observed interference fringes.

If we integrate over the entire k-space or angular range (as one does with normal linear spectroscopy), it is impossible to discriminate between the coherent and incoherent signals. Therefore, a way to discriminate between coherent and incoherent CL is to perform E-k mapping or angle-resolved mapping at tunable frequencies.

Third, our goal is not to judge the beauty and flexibility of all-optical techniques, which we also use in several of our projects. We have developed this CL-based method over several years and intend to exploit its full capabilities and limitations, extending its capabilities to explore the dynamics of solid-state material excitations on several platforms. EELS and CL techniques are linear spectroscopy methods, and several examples of such studies are found in high-impact journals. Our technique combines CL with interferometry, allowing the exploration of quantum superpositions and decoherence phenomena at the nanoscale.

Actions Taken:

To better emphasize the novelty of our scheme, we have added the following sentence to the discussions:

“Our technique transcends the methods available for studying dephasing dynamics by incorporating both luminescence spectroscopy and interferometry in a single scheme. The all-optical analog of this method, which includes photoluminescence (PL) spectroscopy and Mach-Zehnder interferometry, also allows the study of ultrafast dephasing dynamics⁵⁹. However, the sequential CL spectroscopy reported here offers greater flexibility in scanning materials with 1 nm spatial resolution and selecting randomly positioned defects in two-dimensional materials. Moreover, leveraging both coherent and incoherent interactions of electron beams with defects, we are able to recover not only dephasing time, but also population decay in a single experiment.

The interaction of the EDPHS radiation with the sample is similar to the incorporation of coherent radiation in Ramsey-type interferometry schemes preparing the sample in a coherent superposition⁶⁰. The temporal duration within which the system freely evolves, allows for altering the relative phase between the components of the superposition. In contrast with Ramsey-type interferometry though, our second pulse incorporates an electron beam, allowing for both coherent and incoherent interactions, where both the dephasing time and population decay is retrieved.”

Moreover, we added the following section to the Supplementary Note S2:

“To better demonstrate the impact of EDPHS radiation on the emission from defects, we compare the angle-resolved CL patterns of the defect when irradiated only by electron beams (Fig. S6a) to the case when the defect is excited with both electron beams and EDPHS radiation (Fig. S6b). The emission is filtered at a wavelength of 850 ± 25 nm. The distance between the EDPHS and the sample is set to 10 μm .

For the former excitation scheme, where only the electron beam interacts with the defect, the emission does not exhibit coherent radiation properties and lacks a dipolar-like emission pattern as expected. This is due to the incoherent interaction of the electron beam with the defect, involving sequential interactions of secondary or backscattered electrons and random emission in various directions.

Upon illuminating the defect with coherent EDPHS radiation, the angle-resolved CL pattern from the defect exhibits clear interference fringes (Fig. S6b), indicative of coherent radiation properties. EDPHS radiation brings the system into a coherent superposition of the excited states, akin to a $\pi/2$ pulse in optics. This coherent superposition induces linear polarization within the sample, leading to a coherent CL signal. The coherent component of the CL radiation can then interfere with the EDPHS light, resulting in the observed interference fringes.

Fig. S6: Angle-resolved CL pattern of a defect filtered at the wavelength of 850 ± 25 nm. The defect is excited with (a) electron beams at the kinetic energy of 30 keV, and with (b) both electron beams and EDPHS radiation, when the distance between the EDPHS and sample is set to 10 μm .”

**Comment#3:**

The reason I find the approach still appealing is the possibility to measure, in principle, the (sub-) nanometer spatial distribution of a single defect state. As the authors also point out, this is the big advantage of using a scanning electron microscope. However, as long as this advantage is not utilized, I do not recommend the manuscript for publication. The results are also very similar to those presented in [Taleb et al. Nat. Phys. 19, 869–876 (2023)] but on a different sample system and with additional TEM measurements.

Our response:

We thank the reviewer for this comment and for encouraging us to emphasize single emitter coupling. Please kindly note that we do indeed couple to single emitters, since we use focused beams. Also, our theoretical scheme only includes a single 8-level quantum state and does not include coupling to multiple emitters, where the latter, unless being in a superradiance state, cannot lead to the spatio-temporally coherent emission we observe here. In fact, the dynamics we observe here are only purely explained when we couple to single emitters. For more information, please see the new sections added to Supplementary Note S1 and the new Supplementary Note S4, as well as our responses to the first reviewer's comments. We apologize that this aspect was not clarified strongly enough in the manuscript. Now we modified the discussion to explain this aspect in a better way.

Actions Taken:

We have now modified the following paragraph in the discussions:

“While ensuring a long dephasing time for single emitters is important for quantum technologies based on interferometry techniques, coherent phonon polarization in the hBN offers a wealth of possibilities, to enable quantum-sensitive measurements based on novel types of correlations in matter. Coherent phonons lead to an enhanced coupling between emitters, enabling emergent synchronization phenomena. They lead to novel types of superradiance in hBN flakes with a high density of emitters, but need to be further investigated.”

to

“Our results here have been based on the excitation of single emitters with electron beams, agreeing well with our theoretical framework that emphasizes coupling to single emitters. While ensuring a long dephasing time for single emitters is important for quantum technologies based on interferometry techniques, coherent phonon polarization in the hBN offers a wealth of possibilities, to enable quantum-sensitive measurements based on novel types of correlations in matter. Coherent phonons lead to an enhanced coupling between emitters, enabling emergent synchronization phenomena. For this to happen, one could consider coupling the emitters to photonic cavities with their resonant modes taking place within the upper Reststrahlen band (See Supplementary Note 5). They lead to novel types of superradiance in hBN flakes with a high density of emitters, but need to be further investigated. “

**Further comments:**

- The mentioned attosecond resolution is not shown in the data presented in the manuscript. More specifically, only a sub-cycle phase resolution is possible, which only translates to attosecond precision. The authors should be more careful in their formulation.

Our response:

We thank the reviewer for this comment and apologize for the misleading phrasing. We have mentioned attoseconds at two points in the manuscript when we wrote:

“To uncover the population decay T_1 , the delay between the EDPHS and the electron beam arriving at the sample was varied at the steps of only 120 attoseconds, by measuring the integrated CL spectrum over the entire angular range of the emission above the sample, with a collection efficiency of 1.46π steradians (Fig. 4a, top).”

Here, we are referring to the steps of changing the delay between the EDPHS and electron beam excitations, and not to the resolution of our technique. Given the bandwidth of the EDPHS radiation, our temporal resolution is currently 1.4 fs and not less.

And also in the abstract we have mentioned:

“Our results demonstrate the capability of our sequential cathodoluminescence spectroscopy technique to probe the ultrafast dephasing time of single emitters in quantum materials with sub-femtosecond time resolution, heralding access to quantum-path interferences in single emitters coupled to their complex environment.”

We have now removed “sub-“ from the abstract.

Actions Taken:

To better emphasize the temporal resolution of our scheme, we have added the following sentence to the introduction:

“Using a broadband metamaterial-based electron-driven photon source^{37, 38} (EDPHS) that emits sub-cycle photons with a collimated spatial profile and temporal distribution of 1.4 fs, we generate a coherent superposition of phonon states.”

We have also removed “sub-“ from the abstract as mentioned above.

Comment#4:

- The method of CL generation and focusing seems to lead to a not perfectly collimated beam (see Supplement), which should also show chromatic aberrations within the bandwidth of the CL radiation. Therefore, the observed changes in emission and interference intensity as a function of plane distance could also stem from these aberrations.

Our response:

We thank the reviewer for this comment that allows us to better explain the spatial profile of our EDPHS emission. To analyze the field profile at different positions in the EDPHS, we have excited the EDPHS structure with the electron beam at different positions in the supplementary to only explain the sensitivity of the EDPHS radiation to electron-beam impact position. If the electron beam does not excite the EDPHS in the center, the beam will be slightly tilted. This is not the case when we excite the EDPHS at the center, as shown in our previous manuscript (*Nature Physics* 19 (6), 869-876 (2023)) and repeated here for the reviewer's attention:

Figure 3. CL emission profile of the EDPHS when the beam excites the EDPHS in the center. The emission is filtered at $800 \text{ nm} \pm 15 \text{ nm}$.

Furthermore, in all experiments, the electron beam excites the EDPHS structure symmetrically in its center. Such ordered and unambiguous interference maps cannot be explained by random excitation of other structures at tilted angular profiles, that the latter would not lead to any such regular patterns shown in figure 3 of the manuscript.

Response Letter

Reviewer #2 (Remarks to the Author):

I have carefully reviewed the authors' response and the updated manuscript files. They have addressed the reviewers' questions and clarified most of the concerns.

Our Response

We thank the reviewer for the time he/she has invested to review our manuscript. We are pleased that the reviewer finds the current version of the manuscript a significant improvement and believe we have addressed the remaining concerns below.

Comment#1:

While the revised manuscript now specifies where the time-resolved CL measurements were taken and includes a second data point (which shows no effect) in the supplemental material, it remains unclear why the authors have not fully utilized the inherent capabilities of the scanning electron microscope, particularly the scanning.

Our Response

We thank the reviewer for this insightful comment. We would like to confirm that the scanning functionality of our microscope has indeed been employed to locate the emitters through hyperspectral imaging. Almost all electron microscopes scan the sample by adjusting the position or angle of the electron beam over the sample, rather than moving the sample itself. This approach differs from many tip-based and optical microscopy techniques. Consequently, the sample stage of scanning electron microscopes (SEMs) typically has only micrometer-level precision and lacks the accuracy and reproducibility required for high-resolution scanning.

When we use our EDPHS and position it above the sample, the scanning functionality of the SEM cannot be employed, as it would also scan the EDPHS structure. This would result in changes to the angle and profile of the emitted EDPHS radiation, which is undesirable. Therefore, to scan the sample, we would need to precisely control the sample's position relative to the electron beam. Unfortunately, this is not feasible with our current setup, as the SEM sample stage is used to hold the sample itself, while the position of the EDPHS is independently controlled by a piezo positioner. The piezo positioner can move independently of the sample stage but does not allow for precise scanning of the sample.

As a result, for this proof-of-concept experiment, we focused on exploring selected emitters. Nevertheless, to better clarify the role of hyperspectral imaging in verifying the position of the emitters, we have provided statistical data for two hBN flakes below.

**Comment#2:**

At the very least, I would expect measurements from multiple defect centers and clusters, as significant variations in population decay and dephasing times could exist. While this paper is interesting as a proof of concept, a more controlled study with greater statistical significance would be preferable.

The authors have also clarified the distinction between their method and optical spectroscopy techniques. Although their arguments are valid, the method seems most appropriate in the context of spatially resolved measurements, which would reveal the behavior of single or multiple defects. The additional measurement of a defect showing no interference underscores this point. In other words, how can the authors be certain they are investigating a single defect without spatial information?

I believe, with further refinement and a more comprehensive study, this approach could offer significant insights into defect dynamics and might become a valuable tool for the field.

Our Response:

We thank the reviewer for this insightful comment. As the reviewer has noted, our primary objective is to present a proof-of-concept experiment for a newly developed technique that enables the exploration of the dynamics of individual emitters. In the supplementary material, we have already demonstrated various defects and shown that we consistently observe two types of emitters, emitting at 530 nm and 860 nm. This behavior stands in contrast to many published results, which often report a collection of diverse emitters within a single flake, spanning the entire visible range. We attribute this distinct behavior to the high purity of our hBN flakes and the controlled preparation process for exfoliated flakes, which effectively minimizes polymer residues and contamination.

That said, we agree with the reviewer that when emitters are coupled to one another, a variety of dynamic phenomena can be observed. These include phonon-mediated interactions, photon-mediated dipolar interactions, and purely free-induction decay. To gain deeper insights into these interactions, we are currently conducting additional measurements using both photoluminescence spectroscopy and our cathodoluminescence (CL)-based technique. However, exploring these cluster-based dynamics is beyond the scope of the present manuscript.

To address the reviewer's comment, we provide a brief discussion of such cluster-based dynamics below.

Actions Taken:

The following chapter were added to the Supporting Information to show the relaxation dynamics of coupled emitters:

Supplementary Note 6. Excitation of multiple defects and energy-exchange rate between emitters

In the case of excitations of multiple emitters in close proximity, the relaxation dynamics are influenced by the coupling between these emitters. The preparation of hBN thin films via liquid exfoliation enables

the generation of a large number of emitters within a small region. Notably, the sonication duration plays a significant role in determining the number of emitters produced. However, two types of emitters, emitting at wavelengths of 530 nm and 860 nm, are consistently observed. These findings emphasize the critical influence of free-induction decay and emitter-emitter couplings on the decay time and decoherence dynamics of individual emitters, rather than on the specific type of emitter excited.

Hyperspectral cathodoluminescence imaging serves as a powerful tool for determining the positions and numbers of emitters in our hBN films. Two representative hyperspectral images are shown in Fig. S14. Emitters at $\lambda_1 = 860 \text{ nm}$ are less abundant in the flake displayed in Fig. S14a compared to the flake shown in Fig. S14c (see hyperspectral images at $\lambda_1 = 860 \text{ nm}$). In contrast, emitters at $\lambda_4 = 530 \text{ nm}$ are more numerous in this flake.

Figure S14: Statistics of emitters in hBN thin films. (a, c) CL spectra and (b, d) hyperspectral images, at depicted wavelengths of λ_1 to λ_4 , of two hBN films prepared using liquid exfoliation, positioned on the holey carbon substrate. The explored regions are suspended in vacuum. The secondary-electron SEM images of the hBN flakes are provided in the insets of panels (a) and (c).

Emitters at $\lambda_1 = 860 \text{ nm}$ are coupled to coherent phonons, resulting in relaxation dynamics that are significantly influenced when many emitters are positioned in close proximity. This is attributed to direct phonon-mediated energy transfer between emitters, as observed for the flake shown in Figs. S14c and S14d. Conversely, for the flake shown in Fig. S14a, the measured relaxation and decoherence time scales are 510 fs and 180 fs, respectively. These values are comparable to the dynamics of the emitter discussed in the main text.

Using the EDPHS structure shown in Figs. S4 and S5, we examined the relaxation and decoherence dynamics of various hBN flakes. The results presented in Fig. S15 specifically correspond to the flakes shown in Figs. S14a and S14b, focusing on the electron impact position marked as #2 for both flakes.

For the first flake, the population relaxation dynamics were determined by fitting an exponential function to the peak intensities, yielding a relaxation time of approximately 510 fs. In contrast, for the second flake shown in Figs. S14c and S14d, where a higher density of emitters is excited, the coupling between emitters was characterized by fluctuations in the cathodoluminescence (CL) emission intensity as a function of the delay between the EDPHS-generated pulses and the arrival of the electron beam at the sample.

Figure S15: Relaxation dynamics of the emitters at the positions marked by #2 in Fig. S14. (a, b) Population decay demonstrated by acquiring the CL spectra versus the delay between the EDPHS radiation and electrons arriving at the sample. (a) and (b) correspond to the flakes shown in Fig. S14 a and c insets, respectively. (c) Angle-resolved CL maps at depicted delays between electrons and EDPHS radiation, for the flake shown in Figure S14 (c).

The overall relaxation time in the second flake is influenced by free-induction decay between emitters, which also contributes to faster decoherence dynamics and population relaxation. The accelerated decoherence dynamics are evident in the angle-resolved map acquired at a wavelength of $\lambda_1 = 850 \text{ nm} \pm 25 \text{ nm}$ (Fig. S15c), where the visibility of the interference fringes diminishes within 60 fs.

Reviewer #3 (Remarks to the Author):

The only question I have is to ask what is the time resolution (Instrument response function). For such a short time scales, relevant to dephasing the pulse duration of the pump and the probe must be below 100 fs. I haven't found this information.

Our Response:

We thank the reviewer for this comment and apologize for the missing information. The temporal duration of our pump (EDPHS radiation) is 0.9 fs (see Supplementary Notes 2: Experimental Setup, and the bandwidth of the EDPHS radiation). The time resolution of the electron-beam-induced emission is less straightforward, as it depends on many factors. The interaction time of the electron beam with the flake, which occurs on a timescale of only a few attoseconds (depending on the film's thickness), primarily determines the probe's temporal broadening. This broadening is significantly shorter than the pump duration. Therefore, we estimate the temporal resolution of our technique to be within 1 fs. A more accurate estimation is provided by our spectral interferometry measurements (Nature Physics 19, 869–876 (2023)), which, when applied to our present data, **indicate an overall temporal broadening of only 1.5 fs for the EDPHS structure employed here.**

Actions Taken:

We slightly modified our abstract to provide the missing information about the time resolution. The previous sentence:

“Our results demonstrate the capability of our sequential cathodoluminescence spectroscopy technique to probe the ultrafast dephasing time of single emitters in quantum materials with femtosecond time resolution, heralding access to quantum-path interferences in single emitters coupled to their complex environment.”

to the

“Our results demonstrate the capability of our sequential cathodoluminescence spectroscopy technique to probe the ultrafast dephasing time of single emitters in quantum materials with 1.5 femtosecond time resolution, heralding access to quantum-path interferences in single emitters coupled to their complex environment.”